# Stimulating Role of Calcium and Cyclic GMP in Mediating the Effect of Magnetopriming for Alleviation of Salt Stress in Soybean Seedlings

**Sunita Kataria** [1,*] , **Shruti Shukla** [1], **Kanchan Jumrani** [2], **Meeta Jain** [1,*] **and Rekha Gadre** [1]

1  School of Biochemistry, Devi Ahilya Vishwavidyalaya, Khandwa Road, Indore 452001, MP, India
2  Division of Plant Physiology, Indian Institute of Soybean Research, Khandwa Road, Indore 452001, MP, India
*  Correspondence: sunita_kataria@yahoo.com (S.K.); meetajainind@yahoo.com (M.J.)

**Abstract:** This current study examined the role of calcium (Ca) and Cyclic GMP (cGMP) in mitigating the adverse effect of salt stress through magnetopriming of soybean cultivar JS-335 seeds with a static magnetic field (SMF, 200 mT for 1 h). The salt stress (50 mMNaCl) extensively reduced the early seedling growth (64%), vigour Index-I (71%), vigour Index-II (39%), total amylase (59%), protease (63%), and nitrate reductase (NR, 19%) activities in un-primed soybean seedlings. However, magnetopriming and Ca treatment enhanced all of these measured parameters along with remarkable increase in reactive oxygen species (ROS) and nitric oxide (NO) content. The exogenous application of $Ca^{2+}$, cGMP and ROS regulators such as nifedipine ($Ca^{2+}$ channel blocker), EGTA, ethylene glycol-β-amino ethyl ether tetra acetic acid ($Ca^{2+}$chelators), genistein (cGMP blocker), and dimethyl thiourea (DMTU, $H_2O_2$ inhibitor) negatively affects the SMF-induced seedling length, seedling vigour, ROS, NO, and enzyme activities such as protease, total amylase, and NR in soybean seedlings. Results presented by using specific various biochemical inhibitors of Ca, cGMP, or ROS signalling in vivo indicated that Ca and cGMP are also involved with ROS and NO in the signal transduction of magnetic field enthused soybean seed germination and seedling growth under salt stress.

**Keywords:** calcium; cGMP; magnetic field; reactive oxygen species; salt stress

## 1. Introduction

During the last decade the world has experienced an unforeseen change in climate. Salinity is the utmost brutal environmental stressor limiting crop productivity as most of the crops are sensitive to salinity because of high concentrations of salts in the soil [1–3]. Salinization is increasing worldwide and there is decline in the average yields for most crops. Seed germination and emergence are the most vulnerable stages of the plant growth cycle under stress conditions. Various researchers have reported that salt stress may affect germination of seed by generating an osmotic stress thus averting the seed from uptake of water or through the toxic effects of sodium and chloride ions [4–6]. Additionally, salt stress can accelerate various stresses such as oxidative stress, ionic stress, osmotic stress, and hormonal imbalances which affects cell functions, leading to cell damage which ultimately slowdown plant growth [4,5,7].

Plants are able to recognize stimuli and recourse in response to different abiotic environmental stresses by activating defense mechanisms. Various pre-sowing treatments such as magnetopriming which is a non-invasive technique in which dry seeds are treated with a static magnetic field (SMF), improved the seed vigour and germination under various abiotic stress conditions [8–11]. The positive impact of SMF and enrichment of seed germination due to magnetic field (MF) exposure has been observed by various researchers in different crops such as peach (*Prunus persica*) [12], chickpea (*Cicer arietinum*) [13], cucumber (*Cucumis sativus*) [14], lettuce (*Lactuca sativa*) [15], corn (*Zea mays*) [16], tomato (*Solanum lycopersicum*) [17], radish (*Raphanus sativus*) [18], soybean (*Glycine max*), and maize (*Zea*

*mays*) [19]. The effect of magnetic bio-stimulation of seeds under salt stress using SMF was reported in chickpea, soybean, and maize [19–22]. The SMFpre-treatment enhances the tolerance against salt stress by increase in the activity of certain enzymes such as amylase, protease, dehydrogenase, and nitrate reductase [19–22]. Further, Kataria et al. [22] also suggested that magnetoprimed seeds maintain a balance of the plant hormones abscisic acid (ABA), gibberellic acid (GA), and indoleacetic acid (IAA) through the signalling molecule nitric oxide (NO), which helps to counterbalance the negative effects of salinity on seed germination and growth of soybean.

Soybean is the most widely cultivated seed legume and provides food, edible oil, protein concentrate for livestock feeding, constituent in formulated diet of fish and poultry, and a variety of industrial products [23]. Soybean is adapted to a wide range of climate and soil conditions; it is sensitive to various abiotic stresses such as drought, high temperature, metal toxicity, and salinity [8]. Among various abiotic stress, soil salinity is one of the major threat to soybean productivity [1]. Salt stress is answerable for the delayed or reduced germination and vigour of soybean seedlings [1,21,22]. Salinity stress negatively affects the seed germination, plant growth, photosynthesis, biomass accumulation, and ultimately quality and yield of soybean [1,21,22].

One of the intriguing components of magnetic field treatment is that they seemed to upgrade resistance to abiotic [24] or biotic [25] stresses as a consequence of the cell reinforcement reaction actuation. There are reports on the positive effect of magnetopriming on seed germination, early seedling growth, photosynthesis, PS II efficiency, plant growth, and yield on crops such as barley, soybean, wheat, maize, and mungbean under various abiotic stresses including drought, cadmium, salt, and UV-B stress [8,19,21,26–29]. An alleviation strategy, for example magnetopriming, used to safeguard the soybean seedlings from the adverse effects of salinity has been previously identified [21,22]. Further, the role of ROS and NO signalling was reported in the soybean seeds under salt and UV-B stress [21,30]. In addition to ROS and NO, it has also been observed that calcium ions ($Ca^{2+}$) and cyclic Guanosine Monophosphates (cGMPs) are known to play an essential role in the process of signal transduction in plants along with a role in the growth and development [31] under biotic and abiotic stress.

Calcium acts as a universal signal molecule to play pivotal roles in plant growth and development, including cell wall formation [32], osmotic regulation [33], cell division [34], and resistance to biotic and abiotic stresses [35–37]. It has been observed that calcium is an important element for salt tolerance and deliberates a protective role in plants growing under sodic soils. Under stressful conditions, the formation of hydrogen peroxide ($H_2O_2$) and NO increases at the same time and acts together to achieve vital cellular functions [38–40]. Furthermore, in response to abiotic stresses, $H_2O_2$ and NO molecules participate in crosstalk with $Ca^{2+}$ to form a complex signalling network [38,40]. It has been reported that cross-talk between $H_2O_2$ and NO embolden plant tolerance to salinity stress [40,41]. Therefore, our investigation intended to evaluate the role of $Ca^{2+}$ and cGMP along with ROS in the salt stress management in magnetoprimed soybean seeds by considering the seed germination and early growth attributes of seedlings. To accomplish this we used disparate $Ca^{2+}$ and cGMP regulators such as calcium chloride ($CaCl_2$), nifedipine ($Ca^{2+}$ channel blocker), EGTA, i.e., ethylene glycol-β-amino ethyl ether tetra acetic acid ($Ca^{2+}$ chelators), genistein (cGMP blocker), and dimethyl thiourea (DMTU, $H_2O_2$ inhibitor).

## 2. Materials and Methods

### 2.1. Plant Material

Soybean seeds (*Glycine max* [L.] cultivar JS-335) (semi-determinate; having purple flowers, yellow seed coat, and days to maturity is about 100 days) were used for the present study as this variety of soybean seeds are most sensitive to salt stress and provides the best result in response to magnetic field pre-treatment under salt stress [19,21,24,42]. The plant material was obtained from ICAR (Indian Institute of Soybean Research, Khandwa Road, Indore, India).

### 2.2. Treatment under Magnetic Field

Pre-treatment of dry soybean seeds cultivar JS-335 under MF was performed using a fabricated electromagnetic field generator ("AETec" Academy of Embedded Technology, Delhi, India). The exposure was carried out under SMF strength of 200 mT for a time period of about 1 h under ambient conditions of temperature at $25 \pm 5\ °C$ based upon previous studies [19]. During the experiments, un-primed seeds were kept away from the impact of MF. All the experiments were accomplished parallel to the control (un-primed seeds) in the laboratory of Biochemistry department, Devi Ahilya University, Indore (22.7196° N, 75.8577° E), Madhya Pradesh, India.

### 2.3. Screening of Concentration of Different Modulators of cGMP and CaCl$_2$

In order to select the appropriate concentration of modulators to be used during the experiments, screening of these modulators was performed prior to germination of seeds. The modulators of cGMP, CaCl$_2$ and ROS used for the experiment are CaCl$_2$, EGTA, genistein, nifedipine, cGMP analogue, and DMTU. Based on previous research conducted on various concentrations of Ca, cGMP and ROS modulators [43–45] were taken as follows:- CaCl$_2$ (10 mM, 1.0 mM, 0.1 mM), Nef (10 mM, 1.0 mM, 0.1 mM), EGTA (10 mM, 1.0 mM, 0.1 mM), Gen (0.05 mM, 0.1 mM, 1 mM), and DMTU (0.1 mM, 0.05 mM, 1.0 mM). All of these concentrations were used for early seedling growth determination to decide the final concentration of each modulator. The seeds were germinated for five days in appropriate conditions of temperature about $25 \pm 1°C$. After five days (imbibitions for 120 h), different growth parameters such as seedling length, seedling dry and fresh weight, vigour index-I and II were noted and on the basis of results, the final concentrations of the modulators were determined for further experiments as 1mM CaCl$_2$, 1.0 mM Nef, 0.1 mM EGTA, 0.05 mM Gen, and 0.1 mM DMTU.

Germination of the primed and un-primed seeds was completed by surface sterilizing the seeds in 0.01% HgCl$_2$ for 2 min and washing these 3–4 times with distilled water. These sterilized seeds were kept in Petri-plates (15 cm diameter) lined by Whatman filter paper No.1 containing 10 mL of distilled water. For inducing salt stress (saline condition), seeds were placed in Petri-plates (15 cm diameter) containing 10 mL of 50 mM NaCl solution. The primed and un-primed seeds in the absence and presence of NaCl were grown along with the modulators of Ca, cGMP and ROS such as calcium chloride (CaCl$_2$), genistein (Gen), nifedipine(Nef), Ethylene Glycol bis-(β-amino ethyl ether) tetra acetic acid (EGTA) and dimethylthiourea (DMTU).

### 2.4. Early Seedling Growth Characteristics of Seedlings

Primed and un-primed seeds were arbitrarily selected from each treatment in triplicates ($n = 3$) to estimate the early seedling growth parameters such as length of root, shoot and seedling, fresh and dry weight of the soybean seedlings after five days of imbibitions in different modulators of CaCl$_2$, cGMP and ROS. Seedlings were dried in an oven at 70–80 °C and dry weight was taken by the weighing balance. The vigour index of seedlings was estimated as described by Abdul-Baki and Anderson [46].

$$\text{Vigour index I} = \text{Germination \% } \times \text{ Seedling length (Root + Shoot) cm}$$

$$\text{Vigour index II} = \text{Germination \% } \times \text{ Seedling dry weight (Root + Shoot) g}$$

### 2.5. Biochemical Analysis of Seeds

All of the biochemical analysis was carried out in the randomly selected seedlings from each triplicate after five days (120 h seed imbibitions) under different treatments of modulators used under both non-saline and saline conditions.

### 2.5.1. Total Amylase Activity

The total amylase activity was assayed by the Sawhney et al. [47] method. Seedlings (100 mg) were homogenized in 5 mL of 80% chilled acetone, then centrifuged at 15,000 rpm for 10 min at 4 °C. The pellet was dissolved in 10 mL of 0.2 M phosphate buffer (pH 6.4) and again centrifuged at 15,000 rpm for 20 min at 4 °C. The supernatant thus obtained was mixed with 2.0 mL phosphate buffer (pH 6.4), 1.0 mL starch (1%), and incubated for 30 min at room temperature. After incubation, 0.1 N HCl and 0.1 N potassium iodide was added to the mixture and absorbance was recorded at 660 nm. The enzyme activity was expressed as mg starch hydrolysed $mg^{-1}$ protein $h^{-1}$.

### 2.5.2. Protease Activity

Protease activity was determined using the Kunitz method [48]. Seedlings (1.0 g) were crushed in 0.2 M phosphate buffer (pH 7.4) and centrifuged at 13,800 rpm at 4 °C for 30 min. Supernatant (0.5 mL) mixed with 0.5 mL of 1% casein prepared in 0.2 M carbonate buffer and was incubated for 10 min at 37 °C and then 1.0 mL of 10% TCA was added to stop the reaction. This mixture was then centrifuged at 12,000 rpm for 10 min at 4 °C. To the centrifuged solution, 2.5 mL of carbonate buffer along with 0.5 mL of Folin reagent was added and the development of the orange colour was observed and incubated for 30 min at room temperature, then the protein content was measured at 660 nm. The activity was expressed as mg protein hydrolysed $g^{-1}$ fresh weight.

### 2.5.3. Nitrate Reductase Activity

The assay was carried out in seedling tissues based upon the Jaworski method [49]. The chopped seedlings (250 mg) were homogenized in 0.1 M phosphate buffer containing 25% isopropanol and 0.2 M potassium nitrate. The mixture was incubated for 2 h at 30 °C. Next, 0.1% NED and 1.0% sulphanilamide was added to the solution post-incubation, and absorbance was recorded based upon the amount of nitrite formed at 540 nm. The enzyme activity was noted as *n* mole of $NO_2$ $g^{-1}$ FW $h^{-1}$.

### 2.5.4. Estimation of Nitric Oxide

NO estimation was performed by measuring the amount of nitrite formed using the method of Zhou et al. [50] with minor modifications. The seedling tissues (250 mg) of germinated seedlings under different treatments were homogenized in 2.0 mL of 50 mM acetate buffer (50 mM sodium acetate, 50 mM acetic acid, and 4% zinc acetate) (pH 7.6) and centrifuged at 12,000 rpm for 15 min at 4 °C, then the supernatant was saved. The pellet obtained was washed with 0.5 mL acetate buffer and then centrifuged again. Both the supernatants thus obtained were mixed together and filtered using Whatmann filter paper after adding 100 mg of charcoal. The filtrate (1 mL) was mixed with Greiss reagent (1% sulphanilamide and 0.1% NED prepared in 5% $H_2PO_4$) in the ratio of 1:1 and was incubated for 30 min at room temperature, and then the absorbance was recorded at 540 nm. The NO content was expressed as *n* moles $g^{-1}$ FW. The standard curve was prepared using sodium nitrate.

### 2.5.5. Estimation of ROS

Estimation of Superoxide Radical

Superoxide radical estimation was achieved based upon the method given by Chiatanya and Naithani [51] through its ability to reduce Nitrobluetetrazolium chloride (NBT). The seedlings (100 mg) were homogenized in 2.0 mL of pre-chilled 0.2 M phosphate buffer (pH 7.2) containing DTDC (Dithio Carbonic Acid) to inhibit SOD activity. The mixture was centrifuged at 10,000 rpm for 10 min at 4 °C. The supernatant obtained was used for measuring the enzyme activity by recording the absorbance at 540 nm. The activity was measured using extinction coefficient 12.8 $mM^{-1}$ $cm^{-1}$ and represented as µmole of superoxide $g^{-1}$ FW.

Estimation of Hydrogen Peroxide

Hydrogen peroxide content was estimated using Mukherjee and Choudhary [52] by measuring titanium hydro-peroxide complex. The seedlings (500 mg) were homogenized in chilled acetone and filtered by Whatmann No. 1 filter paper. To the homogenate obtained, 2.0 mL titanium reagent (titanium oxide and potassium sulphate processed in concentrated sulphuric acid) and 2.5 mL of ammonium hydroxide solution was added to the filtrate for precipitation of the titanium hydro-peroxide complex. The complex was centrifuged at 13,800 rpm for 15 min at 4 °C. The pellet that was obtained was re-centrifuged after adding 2 M sulphuric acid. After centrifugation, absorbance of the supernatant obtained was noted at 415 nm and the activity was described as μmoles of $H_2O_2$ $g^{-1}$ FW.

*2.6. Statistical Analysis*

These data are expressed as mean $\pm$ S.E. ($n$ = 3) and were analysedusingthe analysis of variance (ANOVA) followed by the post hoc Newman–Keuls Multiple Comparison Test. A ### $p < 0.001$, ## $p < 0.01$, and # $p < 0.05$ indicates the significant difference amongst the un-primed seeds grown in the control with the un-primed seeds grown in different modulators under non-saline or saline conditions and *** $p < 0.001$, **$p < 0.01$, and * $p < 0.05$ indicates the significant difference amongst the magnetoprimed seeds grown in control with the magnetoprimed seeds grown in different modulators under non-saline or saline conditions by using Prism 4 software for Windows, GrafPad Software, San Diego, CA, USA.

## 3. Results and Discussion

Calcium is an essential mineral nutrient for plant growth and development. It plays a pivotal role in maintaining the structural and functional integrity of plant membranes, stabilizing cell walls, controlling ion transport, and regulating ion-exchange behaviour and cell wall enzyme activities [50,51]. We found that 50 mM NaCl remarkably decreased the seedling length (64%), vigour Index-I (71%),vigour Index-II (39%), total amylase (59%), protease (63%), and NR (19%), and ROS content were increased in the seedlings of un-primed seeds while magnetopriming significantly enhanced all of these parameters in non-saline as well as saline conditions (Figures 1–3). To identify the role of $Ca^{2+}$ and cGMP in magnetopriming induced signalling at the alleviation of salt stress in soybeans, we used disparate $Ca^{2+}$ and cGMP regulators such as Nef, nifedipine, 1 mM ($Ca^{2+}$ channel blocker), EGTA, ethylene glycol-β-amino ethyl ether tetra acetic acid-0.1 mM ($Ca^{2+}$chelators), CaCl$_2$ 1 mM ($Ca^{2+}$ analogue), genistein 0.05 mM (cGMP blocker), and dimethyl thiourea 0.1 mM (DMTU). We found that when seedlings were grown in calcium (CaCl$_2$) the seedling length, and vigour Index-I and II were significantly enhanced in SMF-primed seeds in comparison to un-primed seeds under both the non-saline (Figure 1A–C) and saline conditions (Figure 1D–F). However, the extent of promotion was more in seedlings from SMF-primed seeds of Ca (CaCl$_2$ treatment) grown under saline conditions; it showed 24% promotion in vigour Index-I (Figure 1E) and 20% in vigour Index-II as compared to their SMF-primed seedlings grown in controls (Figure 1F). After EGTA, Nef, Gen, and DMTU treatments, all the measured early seedling growth parameters were reduced to a greater extent in SMF-primed seeds in comparison to un-primed seeds under both the conditions (Figure 1A–F).

When compared to un-primed seeds, the SMF treatment caused an increase in $O_2^{\bullet-}$, $H_2O_2$, and NO content in soybean seedlings (Figure 2A–C) under non-saline and saline conditions (Figure 2D–F). After EGTA, Nef, Gen, and DMTU treatment the ROS and NO content was reduced to a greater extent in SMF-primed seeds in comparison to un-primed seeds in both the conditions (Figure 2A–F). In the absence of salt stress, the SMF-induced $O_2^{\bullet-}$content was reduced by DMTU (68%), Nef (57%), Gen (46%), and EGTA (40%) (Figure 2A). Whereas, under salt stress, the $O_2^{\bullet-}$ level was reduced by 66, 61, 54, and 46% by DMTU, Gen, EGTA, and Nef, respectively, in SMF-primed seedlings (Figure 2D). Amongst $Ca^{2+}$ blockers, Nef caused the maximal reduction with 50% and 62.5% in SMF-induced $H_2O_2$ production in non-saline and saline conditions, respectively (Figure 2B,E). cGMP blocker also triggered significant reduction in SMF-induced $H_2O_2$ production, i.e.,

57% and 71%, respectively, in non-saline and saline conditions (Figure 2B,E). The changes in $O_2^{\bullet-}$, $H_2O_2$ and NO levels was not significant in the seedlings of un-primed seeds upon inhibitors treatments except for DMTU (Figure 2A–F).

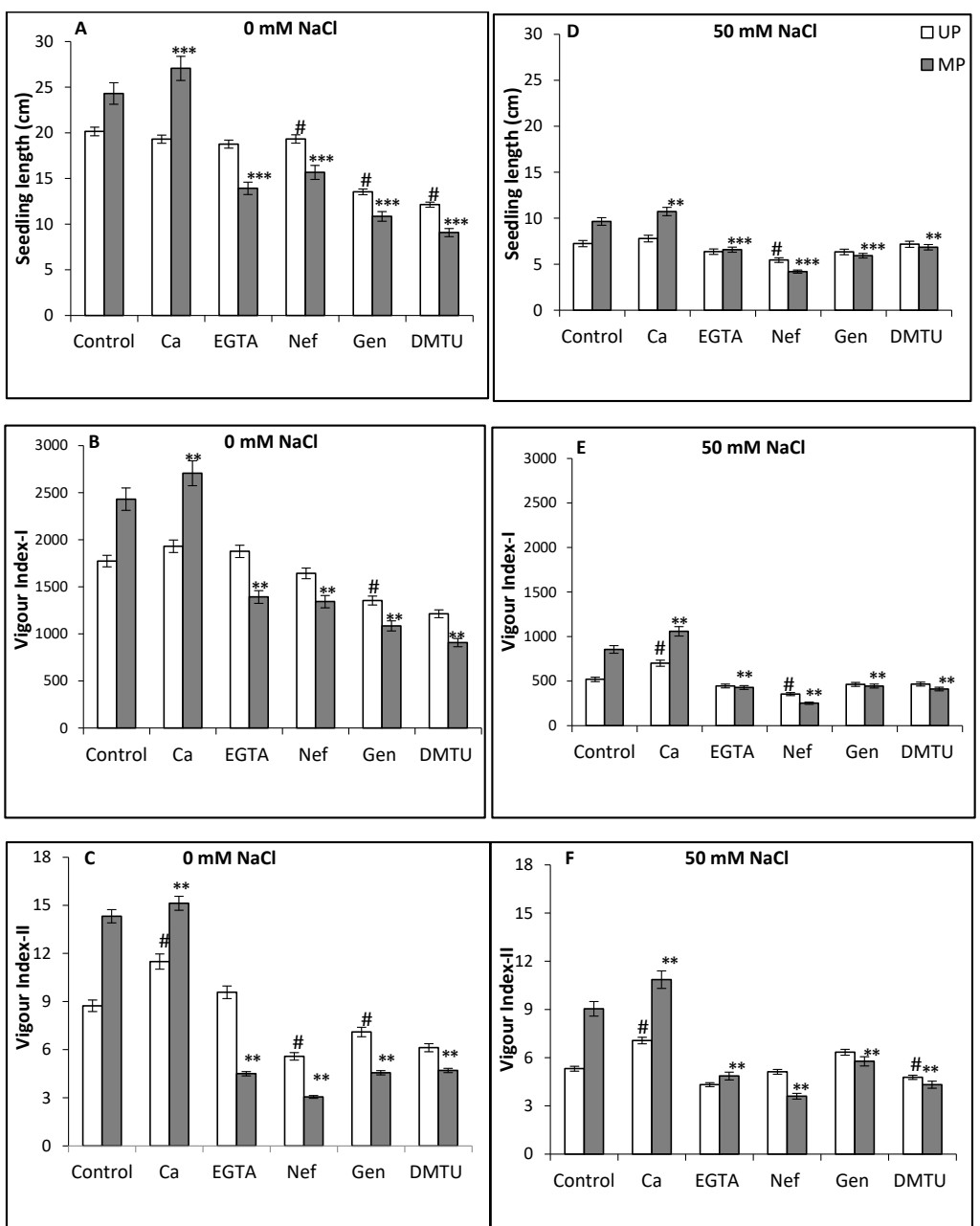

**Figure 1.** Effects of different modulators of calcium, cGMP, and ROS on SMF pre-treatment (200 mT for 1 h) induced seedling length (**A**,**D**), vigour Index-I (**B**,**E**), and vigour Index-II (**C**,**F**) of soybean seedlings, respectively, in non-saline and saline conditions. These data are expressed as mean $\pm$ SE ($n = 3$). $^{\#}$ $p < 0.05$ indicates the significant difference amongst the UP seeds grown in the control with the UP seeds grown in different modulators under non-saline or saline conditions and *** $p < 0.001$, ** $p < 0.01$ indicates the significant difference amongst the MP seeds grown in control with the MP grown in different modulators under non-saline or saline conditions. UP, un-primed seeds and MP, magnetoprimed seeds with SMF. Ca, calcium chloride ($CaCl_2$), 1 mM ($Ca^{2+}$ analogue); EGTA, ethylene glycol-β-amino ethyl ether tetra acetic acid, 0.1 mM; Nef, nifedipine,1 mM ($Ca^{2+}$ channel blocker); Gen, genistein, 0.05 mM (cGMP blocker),and dimethyl thiourea, 0.1 mM (DMTU, $H_2O_2$ inhibitor).

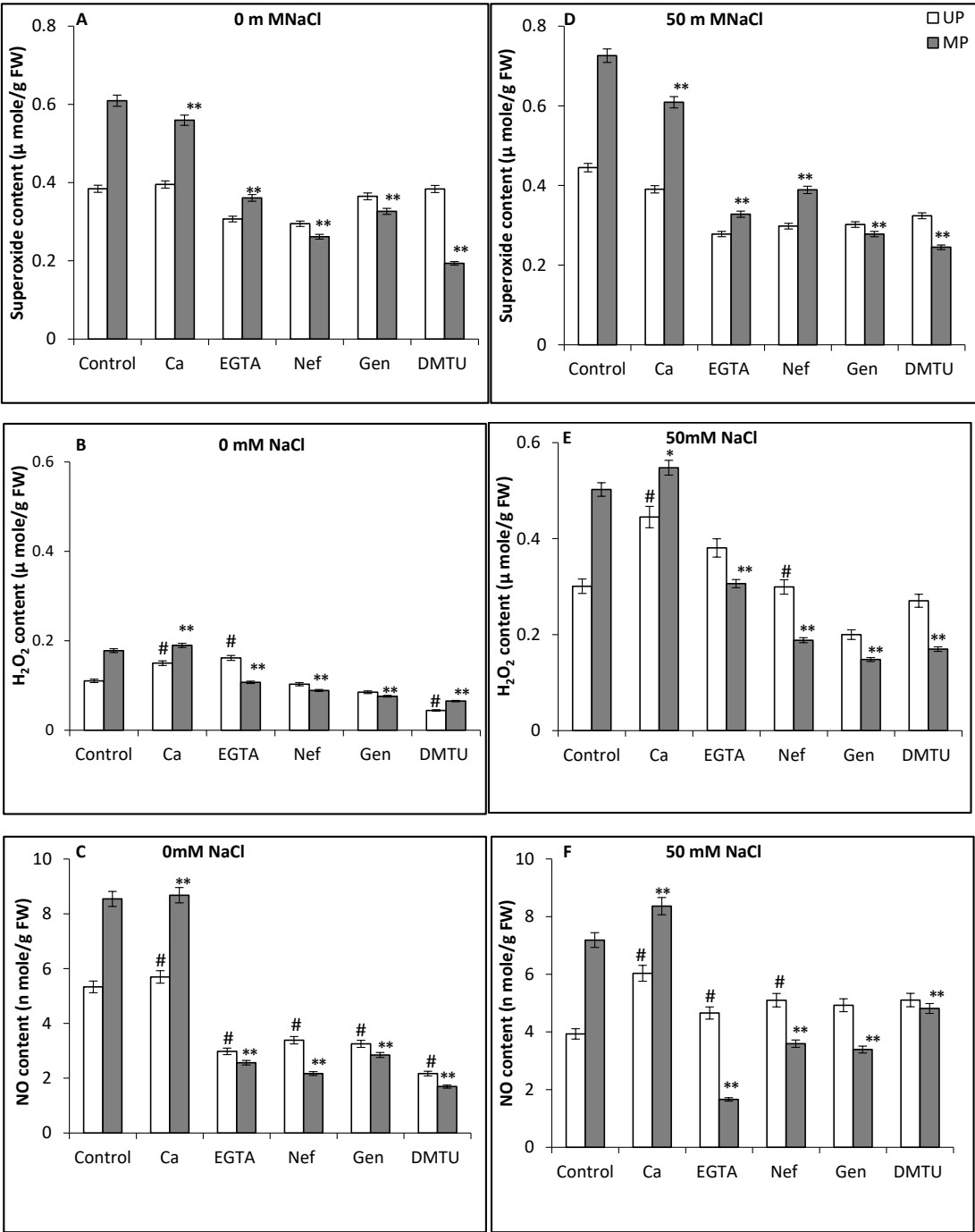

**Figure 2.** Effects of different modulators of calcium and cGMP on SMF pre-treatment (200 mT for 1 h) induced superoxide radical(**A**,**D**), hydrogen peroxide (**B**,**E**) and nitric oxide (**C**,**F**) content in soybean seedlings, respectively, in non-saline and saline conditions. These data are expressed as mean + SE (*n* = 3). [#] *p*< 0.05 indicates the significant difference amongst the UP seeds grown in control with the UP seeds grown in different modulators under non-saline or saline conditions and ** *p* < 0.01, * *p* < 0.05 indicates the significant difference amongst the MP seeds grown in control with the MP grown in different modulators under non-saline or saline conditions. UP, un-primed seeds and MP, magnetoprimed seeds. Ca, calcium chloride (CaCl$_2$), 1 mM (Ca$^{2+}$ analogue); EGTA, ethylene glycol-β-amino ethyl ether tetra acetic acid, 0.1 mM; Nef, nifedipine, 1 mM (Ca$^{2+}$ channel blocker); Gen, genistein, 0.05 mM (cGMP blocker), and dimethyl thiourea, 0.1 mM (DMTU H$_2$O$_2$ inhibitor).

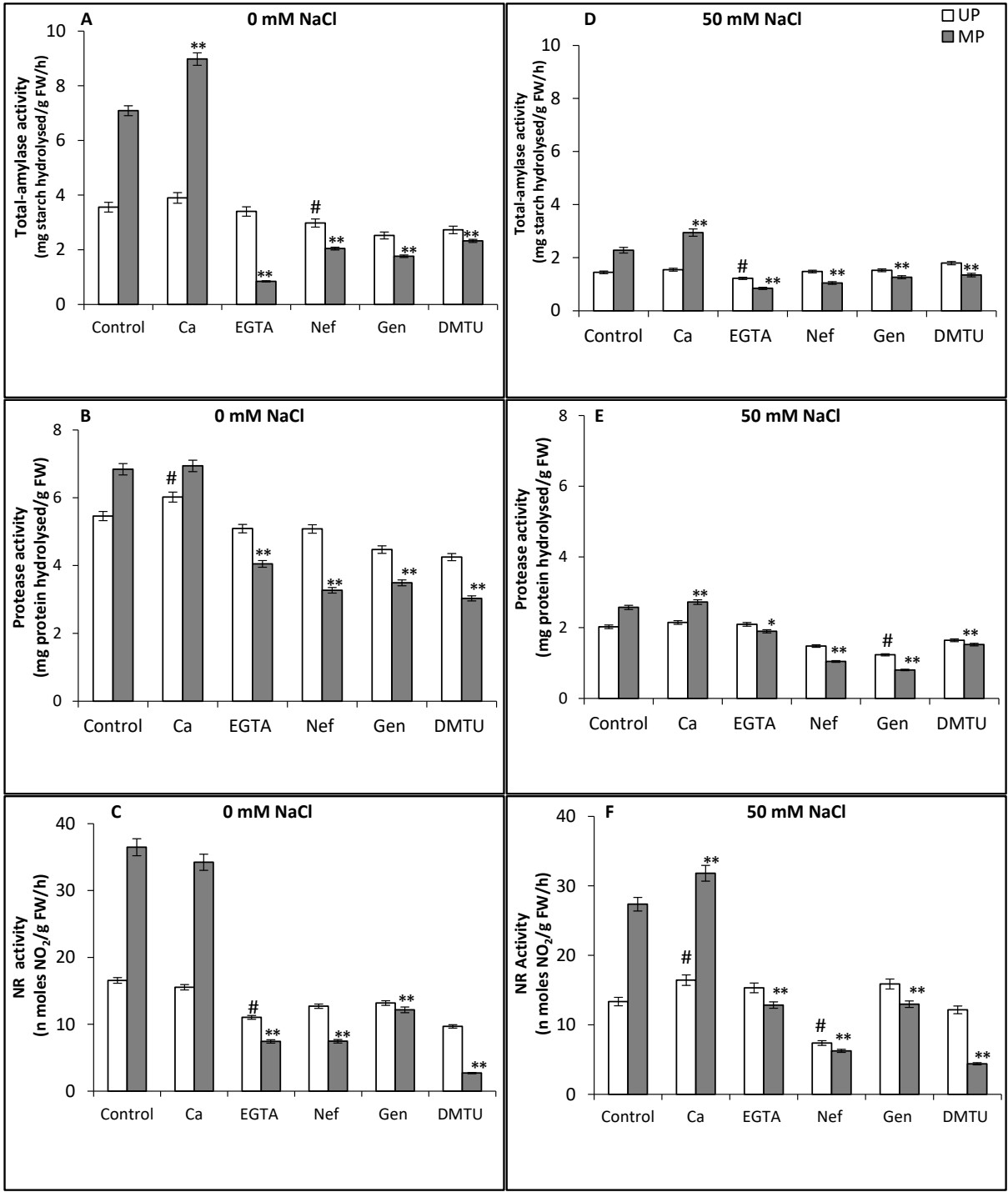

**Figure 3.** Effects of different modulators of calcium and cGMP on SMF pre-treatment (200 mT for 1 h) induced α-amylase (**A**,**D**), protease (**B**,**E**) and nitrate reductase (**C**,**F**) activity in soybean seedlings, respectively, in non-saline and saline conditions. These data are expressed as mean + SE. # $p < 0.05$ indicates the significant difference amongst the UP seeds grown in control with the UP seeds grown in different modulators under non-saline or saline conditions and ** $p < 0.01$, * $p < 0.05$ indicates the significant difference amongst the MP seeds grown in control with the MP grown in different modulators under non-saline or saline conditions. UP, un-primed seeds and MP, magneto-primed seeds. Ca, calcium chloride (CaCl₂), 1 mM (Ca²⁺ analogue); EGTA, ethylene glycol-β-amino ethyl ether tetra acetic acid, 0.1 mM; Nef, nifedipine, 1 mM (Ca²⁺ channel blocker); Gen, genistein, 0.05 mM (cGMP blocker), and dimethyl thiourea, 0.1 mM (DMTU, H₂O₂ inhibitor).

After SMF pre-treatment, the NO content in soybean seedlings was significantly enhanced in the absence (60%) and presence (82%) of salt stress (Figure 2C,F). This increase in NO content was maximally reduced by DMTU (80%), Nef (75%), EGTA (70%), and then Gen (67%) in non-saline conditions (Figure 2C). In presence of salt stress (50 mM NaCl) a reduction in NO content by 76, 50, 53, and 33% after EGTA, Nef, Gen and DMTU treatments, respectively, was observed (Figure 2F). On the other hand, in both saline and non-saline conditions the $CaCl_2$ significantly increased the $H_2O_2$ and NO contents in seedlings of both un-primed and SMF-primed seeds (Figure 2A–F).

$CaCl_2$ accelerated the total amylase activity of SMF-primed seeds by 27% and 29% in the absence and presence of salt stress conditions, respectively (Figure 3D). Maximum reductions of 63% (EGTA), 54% (Nef), 44% (Gen), and 41% (DMTU) were obtained in the total amylase activity of SMF-primed seeds in saline condition (Figure 3D). While un-primed seeds did not show any significant change by the modulators of Ca, cGMP, and ROS in non-saline as well as saline conditions (Figure 3A,D). The protease activity was enhanced by 25% and 27% in soybean seedlings from SMF-primed seeds as compared to un-primed seeds in the absence and presence of salinity, respectively (Figure 3B,E). The protease activity of SMF-primed seeds was declined by all the inhibitors of $Ca^{2+}$, cGMP and ROS (Figure 3B,E). The reduction in protease activity with Nef was 52% and 60% in SMF-primed seeds under non-saline (Figure 3B) and saline conditions, respectively (Figure 3E).

The NR activity in comparison to the un-primed seeds was remarkably enhanced in the soybean seedling during germination in SMF-primed seeds; an increase of 120 and 105% under the absence and presence of salt stress, respectively, was observed (Figure 3C,F). In the presence of salt stress, $Ca^{2+}$ endorsed the activity of NR by 23% in un-primed seeds and 16% in SMF-primed seeds (Figure 3F). While blockers of $Ca^{2+}$ and cGMP/$H_2O_2$ inhibitor DMTU inhibited the SMF-induced NR activity in non-saline as well as saline conditions (Figure 3C,F).

Previous studies have demonstrated that when plants are exposed to salinity stress, their growth is negatively impacted, but an increase in the concentration of $Ca^{2+}$ in the cytosol can help alleviate these negative effects [53–55]. Likewise in our study by adding calcium to the experimental conditions, it showed positive effect on the studied parameters, potentially mitigating the negative impact of salinity in soybean. This is due to the fact that calcium functions as a second messenger in various biological systems, allowing plants to respond to high salt environments by activating a signal transduction pathway involving $Ca^{2+}$ [56–58]. Seed priming can also alter the concentration of $Ca^{2+}$, which triggers a cascade of reactions involving calcium sensors such as calcium-dependent protein kinases (CDPKs) that regulate protein phosphorylation and gene expression during seed germination [59].

In this present study, various inhibitors of $Ca^{2+}$, cGMP, and ROS such as EGTA, Nef, Gen, and DMTU were tested on SMF-induced seedling growth under non-saline and saline conditions. The results showed that all the inhibitors significantly reduced SMF-induced seedling growth and vigour compared to un-primed seeds in both the conditions (Figure 1A–F). These findings suggest that SMF-priming may require the involvement of $Ca^{2+}$ and cGMP in seed germination and seedling growth under salt stress. Previous studies have demonstrated that the addition of $Ca^{2+}$ externally can mitigate the harmful effects of sodium on plant growth under hydroponic conditions, and the application of $Ca(NO_3)_2$ can enhance root nutrient and water uptake [60]. Additionally, research on *Brassica juncea* has shown that $Ca^{2+}$ availability plays a crucial role in ABA-induced inhibition during seed germination, but does not affect the seed germination [43]. It was found that EGTA and nifedipine inhibited seed germination when tried individually and additionally raised the ABA effect, in a synergistic way, when tested simultaneously and concluded that $Ca^{2+}$ is not essential for ABA to cause seed germination [43].

Previous research has shown that treating soybean seeds with SMF can improve the seed germination, which was attributed to an increase in the production of ROS [61]. During seed germination, the generation of free radicals is associated with a shift from a quiescent

state to an active metabolic state, promoting faster germination [62,63]. High levels of ROS can act as signalling molecules, promoting rapid axis growth and mobilisation of reserve materials [64]. Additionally, ROS production during seed germination helps with processes such as cell wall elongation, endosperm weakening, redox regulation, hormone and $Ca^{2+}$ signalling, protection against pathogens, and gene expression [65]. Our results showed that magnetoprimed seeds had a significant increase in ROS production, including $O_2^{\bullet-}$, $H_2O_2$, and NO content under both saline and non-saline conditions. Furthermore, the addition of $Ca^{2+}$ further enhanced SMF-induced $H_2O_2$ and NO production, while the use of inhibitors such as EGTA, Nef, and DMTU significantly reduced ROS and NO production in both saline and non-saline conditions (Figure 2).

In addition, we found that magnetoprimed seeds exhibited higher levels of protease, amylase, and nitrate reductase activity compared to un-primed seeds under both saline and non-saline conditions (Figure 3A–F). Furthermore, the addition of $Ca^{2+}$ further enhanced SMF-induced enzyme activities, while the use of inhibitors such as EGTA, Nef, and DMTU reduced the enzyme activities in both saline and non-saline conditions (Figure 3A–F). During seed germination, $\alpha$ and $\beta$-amylase enzymes results in the breakdown of stored carbohydrate in to the monosaccharides which are utilised by the growing seedlings. Previously, it has been reported that in soybean, chickpea, and maize under salinity stress there is increased rate of seed germination in magnetoprimed seeds because of higher activity of amylase, protease, and nitrate reductase [19,20,42,66]. Results clearly indicated that all the inhibitors of Ca/cGMP/ROS declined the NR activity under saline and non-saline conditions, and it was maximally repressed by Nif and DMTU in un-primed and SMF-primed seeds under both conditions (Figure 3C,F).

Our results suggest that SMF may activate $Ca^{2+}$ receptors and target proteins to enhance $[Ca^{2+}]$ cytosolic level and $Ca^{2+}$ during seed germination may persuade the generation of $H_2O_2$ in salinity stress. Li et al. [67] reported that in *Arabidopsis thaliana* roots salt stress induces $H_2O_2$ accumulation in $Ca^{2+}$-dependent salt resistance pathway. It is also revealed that $Ca^{2+}$ signalling induced NO accumulation through inducing $H_2O_2$ generation during stomatal closure of guard cells in *Arabidopsis* [68]. Moreover, a correlationship among $H_2O_2$, calcium-sensing receptor (CAS) and NO was found in $Ca^{2+}$-dependent guard cell signalling [69]. In the present study, we also found that the SMF-induced NO generation was suppressed by $Ca^{2+}$ channel blockers (EGTA/Nef) and cGMP blockers (Genestein) indicating that $Ca^{2+}$/cGMP may mediate the effect of SMF on NO production. It was also shown that, $Ca^{2+}$ released through various type of $Ca^{2+}$ channels was activated by NO and $H_2O_2$ [70]. NO is a signalling molecule, with transduction through a cGMP-independent or cGMP-dependent pathway. In the cGMP-dependent pathway, NO signalling involves $Ca^{2+}$,cGMP, cADPR, and protein kinases [71].

It has been demonstrated that the $Ca^{2+}$ is the downstream targets of NO and it may act through cGMP and cADPR to control intra-cellular $Ca^{2+}$ channels to increase free cytosolic calcium [72,73]. It was shown that NO was able to activate both intra-cellular and plasma membrane $Ca^{2+}$ channels via cascades of reaction involving plasma membrane depolarization, CADPR, and protein kinase [72–74].The role of cGMP is strongly associated with the NO signal cascade in various physiological processes, such as seed germination and gibberellic acid(GA) induced $\alpha$-amylase production [44,74,75]. Similarly, previously we found that magnetopriming-induced salt tolerance by NO signalling further activated GA synthesis and reduced the ABA content in soybean seeds during seed germination [22]. Thus, SMF exposure appears to stimulate a cascade of signalling events that culminate in the activation of key enzymes involved in seed germination and seedling growth. These findings suggest that SMF treatment may be a promising strategy for improving plant growth and yield in salt-affected soils. However, further research is needed to confirm these findings and optimize the SMF treatment conditions for maximum benefit.

## 4. Conclusions

In conclusion, it appears that exposure to SMF can activate calcium receptors and target proteins, leading to enhanced calcium and cGMP signalling. This signalling pathway may trigger the production of $H_2O_2$, $O_2^-$, and NO, which in turn may activate total amylase, protease, and nitrate reductase activities. Under salt stress conditions, the activation of these enzymes by SMF-priming may play an important role in accelerating seed germination and promoting the seedling growth in soybean plants. The addition of $Ca^{2+}$ further enhanced SMF-induced growth and enzyme activities, while the use of inhibitors such as EGTA, Nef, Gen, and DMTU reduced all the parameters in both saline and non-saline conditions. Indeed, the increased $Ca^{2+}$ resulting from SMF exposure may trigger the $Ca^{2+}$-mediated activation of the SOS signalling pathway, which plays a crucial role in maintaining ion homeostasis in plants under salt stress. These indicate that the SMF-induced tolerance against salt stress in soybean seedlings may be mediated, at least in part, by a complex interplay among $Ca^{2+}$, cGMP, and ROS signalling pathways. Further studies are needed to elucidate the precise mechanisms underlying this interplay and to identify specific molecular targets that can be manipulated to enhance plant tolerance to salt stress. This study could ultimately help to develop strategies for enhancing plant tolerance to abiotic stress, which is essential for sustainable agriculture in the face of climate change.

**Author Contributions:** S.K. and M.J. designed the experiments; S.S. and S.K. performed the experiments; S.K. and S.S. analysed the data; S.K. and M.J. wrote the manuscript; S.K., M.J., K.J., R.G. and S.S. finalised the manuscript in its final form. All authors have read and agreed to the published version of the manuscript.

**Funding:** This research work received financial support from the Department of Science Technology (SR/WOS-A/LS-17/2017-G) to S. Kataria.

**Institutional Review Board Statement:** Not applicable.

**Informed Consent Statement:** Not applicable.

**Data Availability Statement:** Data can be obtained on request from the authors.

**Conflicts of Interest:** The authors declare no conflict of interest.

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
