# Peer review of "Stimulating Role of Calcium and Cyclic GMP in Mediating the Effect of Magnetopriming for Alleviation of Salt Stress in Soybean Seedlings"

_2674-1024, doi:10.3390/seeds2020018_

Round 1

Reviewer 1 Report (Previous Reviewer 3)

The authors have now significantly improved the quality of this manuscript and I recommend acceptance of the manuscript after minute changes as suggested below!

1: Some significant numerical results in the Abstract would be a great addition. 

2: Scientific Names of plants should be used instead of common names.

3: In the introduction, the authors could shed more light on the effect of used amendments against salinity stress in various crops instead of simply listing one line for each.

4: The GPS coordinates of this study can be added in the methods section.

5: I am unable to see the "numbering of each heading and sub-heading.

6: The statistical results seem better now. 

7: Where is "the conflict statement?

8: The conclusion should be under a separate heading I believe. 

Dear Authors;

I appreciate the quality of the English of this manuscript as it seems the authors have paid attention to the English corrections.

Author Response

Reviewer 1

 Specific comments

The authors have now significantly improved the quality of this manuscript and I recommend acceptance of the manuscript after minute changes as suggested below!

Answer:  We thank the reviewer for providing useful suggestions that have helped us to further improve this manuscript. Following are the responses to the comments of the reviewer.

Detailed comments and suggestions:

Comment-1 Some significant numerical results in the Abstract would be a great addition. 

Answer:  It has been added in the abstract.

Comment-2 Scientific Names of plants should be used instead of common names.

Answer:  Scientific names of plants has been added instead of common names

Comment-3 In the introduction, the authors could shed more light on the effect of used amendment against salinity stress in various crops instead of simply listing one line for each.

Answer:  It has been now included in the introduction of the manuscript.

Comment-4 The GPS coordinates of this study can be added in the methods section.

Answer:  The GPS coordination of this study has been added in the methods section.

Comment-5 I am unable to see the "numbering of each heading and sub-heading.

Answer:  numbering of each heading and sub-heading has been added in the whole manuscript.

Comment-6   The statistical results seem better now. 

Answer: Thanks for your suggestions that have helped us to further improve this manuscript.

Comment-7 Where is "the conflict statement?

Answer: The conflict statement has been added in the manuscript.

Comment-8: The conclusion should be under a separate heading I believe. 

Answer: The conclusion has been added in separate heading.

Comment-9 I appreciate the quality of the English of this manuscript as it seems the authors have paid attention to the English corrections.

Answer: Thanks a lot for your suggestions that have helped us to further improve this manuscript.

Reviewer 2 Report (Previous Reviewer 1)

The authors of the manuscript took up an interesting topic related to salt stress and the important plant soybean. The research performed and described, as well as the graphic material, are correct and the manuscript as a whole deserves to be published.

Author Response

Reviewer 2

Specific comments

The authors of the manuscript took up an interesting topic related to salt stress and the important plant soybean. The research performed and described, as well as the graphic material, are correct and the manuscript as a whole deserves to be published.

Answer:  We thank the reviewer for providing useful suggestions that have helped us to further improve this manuscript.

This manuscript is a resubmission of an earlier submission. The following is a list of the peer review reports and author responses from that submission.

Round 1

Reviewer 1 Report

Main comments:

The manuscript submitted for review addresses the issue of alleviating salt stress for germinating soybeans. Due to the results not yet fully discovered and extensive research, it may seem very interesting.

However, the text style and deficiencies, especially in the Materials and Methods chapter, require improvement.

Detailed comments and suggestions:

Abstract

Line 11: soybean seeds, cv.? (fill in what was the cultivar of soybeans?)

Keywords: commas should be placed between words

Introduction

Why does italic appear for Seed germination, stages of plant growth, and other expressions?

Line 50-52: This sentence is not correct, it needs to be improved, made more specific.

The text should be analyzed and the fragments concerning stress should be moved before the descriptions of the magnetic field. The text (Line 46-55) on soybeans should be moved before the text describing the effects of the magnetic field (Line 38-45)

Materials and Methods

Line 84-85: There should be a description of the variety, earliness group it belongs to, seed size (1000 seed weight) cv. instead of var., as this cultivar.

Line 91: Were the seeds dry or moist?

Line 112: What was the size of the Petri dishes?, how much water was used for one dish. How many seeds were sown per plate?

Line 123 and 124: Incomprehensible, it is about the length of the roots and the length of the hypocotyl of seedlings, the size of the cotyledons. Why is it written about seeds? What day were these measurements taken?

Line 130: On what day after sowing the seeds was the test material collected?

Results and Discussion

Line 216: Figure 1 and Fig. 2 and Fig.3. Abbreviations are missing in the figures description (EGTA, Nef, Gen, 8 Br G, DMTU)

Reviewer 2 Report

Manuscript entitled “Stimulating role of Calcium and Cyclic GMP in mediating the 2 effect of magnetopriming for alleviation of salt stress in soy-3 bean” by Sunita Kataria et all, address the study of role of calcium and cyclic GMP for alleviation of salt stress in Glycine max.

The work is interesting but, in my opinion, the manuscript presents some issues mainly defects of form and other things, that need to be checked to make this work suitable for publication.

Line 5. Please correct “Sunita Kataria1*.Shruti Shukla1.Kanchan Jumrani2. Meeta Jain*1.Rekha Gadre1.” with “Sunita Kataria 1,*, Shruti Shukla 1, Kanchan Jumrani 2, Meeta Jain 1,*, Rekha Gadre 1”.

Lines 8 and 9. Please correct “Correspondence: Dr Sunita Kataria and Dr Meeta Jain,School of Biochemistry, Devi AhilyaVishwavidya-8 laya, Indore-452001, M.P., India,Email Id:sunita_kataria@yahoo.com; meetajainind@yahoo.com” with “Correspondence:  sunita_kataria@yahoo.com (S.K.); meetajainind@yahoo.com (M.J.)”.

Line 23. Please correct “Keywords: Calcium.cGMP.magnetic field. reactive oxygen species.salt stress” with “Keywords: calcium; cGMP; magnetic field; reactive oxygen species; salt stress”

Lines 30-32. Please remove italics

Line 41. Please check “[8-11].The”. Please correct with “[8-11]. The”. Please check the entire manuscript, as it is full of this type of error (e.g. 54, 55, 56, 63, 85, 93, etc.)

Line 46. Please correct “[19, 20-22].” with “[19,20-22].”.

Line 61. Please correct “[8,19, 21, 26-29].” with “[8,19,21,26-29].”

Line 87. Please correct “[19, 21, 24,42].” with “[19,21,24,42].”

Line 318. Please correct “[19, 20, 42, 64].Results” with “[19,20,42,64]. Results”

Lines 126. Please, “Arabidopsis thaliana” and “Arabidopsis” should be in italics.

Lines 146-147. Please correct “salt overly sensitive (SOS) pathway” with “SOS pathway”.

Line 214. K+. The “+” should be in superscript.

Line 299. Please correct “reactive oxygen species (ROS) and” with “ROS and”

Lines 336-337. Please correct “increases CATALASE2 (CAT2) activity” with “increases CAT2 activity”

Line 340. Please correct “CatC, a CATALASE (CAT) is phosphorylated” with “CatC, a CAT is phosphorylated”

Lines 101 and 117. “e.g. DMTU”. Please review all abbreviations in the text

Lines 93,101-104, 114, 119-120, etc. Please the units should appear separated from the numbers.

Line 178. Please check “mM-1 cm-1 M”. Are these units correct?

All text. Please correct “Fig.” with “Figure”

All figure legends. Please add the abbreviations that appear in the panels (EGTA, Nef, Gen,8 Br G, DMTU, UT, MT). Add de “n”, add de statistic analysis, add dthe P value, etc.

Please check the panel  and B in Figure 1 (the numbers are not visible on the Y axis).

Regarding section References

The references list has numerous format mistakes

Reference 1. Please correct “S, K.” with “S,K.”. Revise all references

Reference 2. Please correct “A.;Ahmar” with “A.; Ahmar”. Revise all references

Reference 2. The volume  and journal articles should be in italics. Revise all references

According to the “Instructions for Authors” (Journal Articles - Author 1, A.B.; Author 2, C.D. Title of the article. Abbreviated Journal Name Year, Volume, page range.) the references list has numerous format mistakes. The correct form of Abbreviated Journal Name is for example in reference 44 “J. Plant Physiol.” and not “J Plant Physiol”.

Please revise all references.

The authors should also show interest in the formal aspects of the manuscript.

Reviewer 3 Report

The authors have examined the role of calcium and Cyclic GMP (cGMP) in mitigating the 10 adverse effect of salt stress through magnetopriming of soybean seeds. I have critically reviewed the manuscript and found flaws in the experimental results and objectives of the study. Therefore, I recommend the rejection of this manuscript. My concerns are below!

1: Why did the authors choose 0 mM NaCl and 50 mM NaCl salinity levels? The results clearly says that 50 mM NaCl stress did not affect the measured parameters as compared to the 0 mM NaCl level. So, if there is no effect of salinity stress on soy-bean traits, then how do the authors expect that calcium and Cyclic GMP could be effective against salinity stress?

2: I was looking for UT and MT in the whole manuscript, and could not find information on these abbreviations used in the figures.

3: The scientific language is full of flaws and needs to be revised by a native speaker.

4: There has no basis provided for the selection of the dosages used in the current experiment.